# The Impact of Epigallocatechin Gallate and Coconut Oil Treatment on Cortisol Activity and Depression in Multiple Sclerosis Patients

**DOI:** 10.3390/life11040353

**Published:** 2021-04-17

**Authors:** Jose Luis Platero, María Cuerda-Ballester, David Sancho-Cantus, María Benlloch, Jose Joaquin Ceron, Camila Peres Rubio, María Pilar García-Pardo, María Mar López-Rodríguez, Jose Enrique de la Rubia Ortí

**Affiliations:** 1Doctoral Degree School, Catholic University of Valencia San Vicente Mártir, 46001 Valencia, Spain; joseluis.platero@mail.ucv.es (J.L.P.); m.cuerda@mail.ucv.es (M.C.-B.); 2Department of Nursing, Catholic University of Valencia San Vicente Mártir, 46001 Valencia, Spain; maria.benlloch@ucv.es (M.B.); joseenrique.delarubi@ucv.es (J.E.d.l.R.O.); 3Interdisciplinary Laboratory of Clinical Analysis, Campus of Excellence Mare Nostrum, University of Murcia, 30100 Murcia, Spain; jjceron@um.es (J.J.C.); camila.peres@um.es (C.P.R.); 4María Pilar García Pardo, Department of Psychology and Sociology, University of Zaragoza, Campus Teruel, 44003 Teruel, Spain; magarpar@unizar.es; 5Department of Nursing, Physiotherapy and Medicine, University of Almería, 04120 Almería, Spain; mlr295@ual.es

**Keywords:** multiple sclerosis, epigallocatechin gallate, coconut oil, depression, fat

## Abstract

(1) Background: Multiple sclerosis (MS) is pathogenically characterized by high oxidative stress and symptomatically by progressive muscle loss and increased body fat associated with the presence of depression. Epigallocatechin gallate (EGCG) (particularly present in green tea) and ketone bodies (in particular beta-hydroxybutyrate (BHB)), whose main source is coconut oil, have shown emotional benefits and body fat loss. The aim of this study was to assess the impact of EGCG and coconut oil on cortisol activity related to fat loss and depression in MS patients. (2) Methods: The study involved 51 MS patients who were randomly divided into an intervention group or a control group. The intervention group received 800 mg of EGCG and 60 mL of coconut oil, which were included in their daily diet for four months. The control group received placebo and all patients followed an isocaloric diet. A blood sample was collected before and after the four-month period, and levels of cortisol, albumin and BHB were measured in serum. In addition, immediately before and after the intervention, anthropometric variables were measured: waist-to-hip ratio (WHR), body fat mass percentage, fat weight, total weight, and muscle mass percentage. Depression was assessed with the Beck Depression Inventory II (BDI-II). (3) Results: No significant changes were obtained in cortisol levels in any of the groups, and there was a significant increase in albumin in the blood of the intervention group only that could lead to a decrease in serum free cortisol. In addition, it was observed a significant decrease in levels of depression and abdominal fat. (4) Conclusions: EGCG combined with coconut oil increase the concentration of albumin in blood and produce less depression in MS patients.

## 1. Introduction

Multiple sclerosis (MS) is an autoimmune disease of an inflammatory nature characterized by the progressive loss of myelin that covers the axon of neurons [1]. One of the pathogenic mechanisms of the disease is due to high oxidative stress, mainly based on generating mitochondrial reactive oxygen species (mtROS). They stimulate NLRP3 inflammasomes which, when acting on the mitochondria, promote local inflammatory lesions. This then leads to mitochondrial disorders, resulting in neurodegeneration and even higher levels of oxidative stress [2]. In turn, an increase in oxidative stress is associated with a deregulation of the hypothalamus–hypophysis–adrenal gland axis (HHA) [3]. This deregulation is characterized by a hyperactivity of HHA, which increases cortisol levels which, in turn, are related to the presence of depression. This has been evidenced mainly because, in people with depression, the following are evident: the secretion of cortisol and adrenocorticotropic hormone (ACTH), responsible for the stimulation of cortisol synthesis in the adrenal cortex, increases, which is also reflected in high rates of urinary cortisol production; corticotropin-releasing stimulating hormone (CRH) levels in cerebrospinal fluid are elevated; there are increases the number of CRH-secreting neurons in the limbic regions of the brain; and the number of CRH binding sites in the frontal cortex is reduced as a consequence of the increase in the CRH concentration [4,5].

Depression is present in most MS patients [6], playing a remarkably relevant role in disease progression [7]. However, in MS, cortisol’s role in the presence of depression and progression of the disease is not clear, since, on the one hand, hyperactivity of HHA in some patients has been associated with a slower progression of the disease, given the glucocorticoids immunomodulating activity [8]; whereas, on the other hand, excess cortisol also leads to progressive damage in mitochondrial activity, causing even higher levels of oxidative stress [9,10]. This is why the relation of cortisol with the presence of depression in MS patients could be due to other mechanisms. Among these, it should be noted that the presence of abdominal fat—since chronically high levels of cortisol have been associated with depression through abdominal fat presence [11]—is present in the majority of these patients [12], and is linked to a worse prognosis [13]. Specifically, it is the active cortisol in free form which goes through the cellular membrane, the one positively correlated with fat percentage [14], and mainly from the abdominal region due to a higher expression of 11β-hydroxysteroid dehydrogenase type 1 (11β-HSD1) in this region [15]. Since it is a molecule of a fat-soluble nature, levels of free cortisol (approximately 6%) [16] depend globulin (GBP, globulin binding protein) which mostly carries it [17], as well as and the albumin which carries it by 14% [18]. This is why treatments that alter blood albumin levels could vary active cortisol levels. All these processes are collected in Figure 1. 

In order to treat depression, which is on the rise in these types of patients, usually antidepressants are being given [19]. However, one of the main problems that the intake of antidepressants in MS presents is the fact that some of the side effects of these drugs can worsen deficits related to the course of the disease. For instance, selective serotonin reuptake inhibitors (SSRI) have been seen to cause or exacerbate the sexual dysfunctions or decreased sexual desires present in many MS patients [20,21]. In this sense, certain antioxidants and nutrients could be promising therapeutic alternatives, and show favorable results due to their anti-inflammatory and neuroprotective activities. Both green tea, characterized by its high levels of polyphenols, especially epigallocatechin gallate (EGCG), as well as nutrients that increase ketone bodies in the blood, such as coconut oil, show great benefits when treating neurodegenerative diseases, as mitochondrial activity is improved [22,23]. Specifically, in the treatment of depression, both EGCG and ketogenic diets have been shown to be effective [24,25,26,27]. This antidepressant effect could, in turn, be linked to the lipolytic activity that shows (both EGCG and ketogenic diets), especially at an abdominal level [28,29,30,31,32].

Taking all the aforementioned into account, the aim of this study was to establish the impact of the administration of EGCG, supplemented with coconut oil in cortisol activity, and its relations with possible improvements in the perception of depression associated with abdominal fat loss.

## 2. Materials and Methods

A prospective, mixed, and experimental pilot study was conducted. The clinical trial ID for this study is NCT03740295.

### 2.1. Subjects

The population sample was obtained by contacting the main MS associations in Spain, which informed their members on the nature of the research. The following selection criteria were applied to the 67 people interested in participating in the study: patients over 18 years of age diagnosed with MS at least 6 months ago and treated with glatiramer acetate and interferon beta (a common treatment for the disease). Moreover, the exclusion criteria included: pregnant or breastfeeding women; patients with tracheotomy; stoma or with short bowel syndrome; patients with dementia; evidence of alcohol or drug abuse; patients with myocardial infarction; heart failure; cardiac dysrhythmia; symptoms of angina or other heart conditions; patients with kidney conditions with creatinine levels two times higher than normal; patients with elevated liver markers three times higher than normal or with chronic liver disease; patients with hyperthyroidism; patients with acromegaly; patients with polycystic ovary syndrome; or MS patients who were included in other studies with experimental drugs or treatment. A CONSORT diagram is attached that describes the flow of participants, from joining the study to analyzing statistical data (Figure 2), where it can be noted that 16 of the patients who wanted to participate in the study were excluded because they did not meet some of the selection criteria; specifically 10 were taking another type of medication that was not glatiramer acetate or interferon beta, 4 were diagnosed less than 6 months before, and 2 were younger than 18.

### 2.2. Statistical Analysis

The SPSS v.23 (IBM Corporation, Armonk, NY, USA) tool was used for the statistical analysis. The first step involved estimating the distribution of the variables investigated through statistical methods in order to assess normality; this also included the use of the Kolmogorov–Smirnov Test. A non-normal distribution of all the scale variables studied in this analysis was demonstrated. Therefore, a Mann–Whitney U test and Wilcoxon signed-rank test were used to assess the inter-group and pre–post differences, respectively. A chi-square test was used to analyze categorical data. A *p*-value below 0.05 was considered significant. Data are presented as mean ± standard deviation, or the number of patients and the percentage representing this number for each group (control and intervention groups).

### 2.3. Procedure

All participants received specific information on the nature and aim of the study. After the final sample was selected, they were provided with instructions indicating not to change the established diet for each case (depending on which group they were assigned to), as well as to take the capsules every day at the agreed times over the 4-month duration of the intervention. Weekly telephone calls were made to all patients to verify whether they were complying with the treatment. They were also asked about any doubts and issues about following the intervention.

### 2.4. Intervention

A final sample of 51 MS patients was obtained after the selection criteria had been applied. Participants were randomly divided into the intervention and control groups. Randomization of patients to either group was performed without stratification by drawing consecutively numbered sealed envelopes. The intervention group received an isocaloric diet for 4 months (adapted to each patient’s characteristics and divided into 5 meals a day: breakfast, mid-morning snack, lunch, afternoon snack and dinner), enriched with 60 mL of extra virgin coconut oil divided into 2 equal intakes (30 mL for breakfast and 30 mL for lunch), and supplemented with 800 mg of EGCG, administered in 2 capsules of 400 mg to be taken twice a day (one capsule in the morning and another in the afternoon). 

On the other hand, the same isocaloric diet was also followed by the control group for 4 months, with the exception of the coconut oil that was replaced with other lipids, such as olive oil, sunflower oil, fish and nuts. This ensured that a state of ketosis did not occur in the control group, as the percentages and the amounts of macronutrients were not characteristic of a ketogenic diet [33]. Placebo (capsules containing microcrystalline cellulose, matching in size and color) was also administered to the control group for the same 4 months, and they followed the same instructions as the intervention group. The basal diet followed by both groups included the following percentage distribution of the 3 main macronutrients, with respect to the total caloric value: 20% proteins, 40% carbohydrates and 40% Mediterranean lipids.

### 2.5. Measurements

The following measurements were taken before and after the 4-month intervention, in the same conditions and at the same time. In terms of the specific case of the questionnaire establishing levels of depression, it was carried out by the same neurologist assigned to each patient before the study. 

Body composition: Each subject received detailed information on the procedure that was to be followed in the assessment in terms of how the anthropometric variables were measured. Informed verbal and written consent were requested. The measurements were taken by an ISAK (The International Society for the Advancement of Kinanthropometry) level 3 certified anthropometrist, in line with the protocol established by the Society [34]. The validated anthropometric material used was as follows: a portable clinical scale, SECA model, with a 150–200 kg capacity and 100 g precision; height rod, SECA model, 220 Hamburg, Germany, with a 0.1 cm precision; metal, inextensible and narrow anthropometric tape, model Lufkin W606PM, with 0.2 mm precision; a mechanical skinfold caliper, model Holtain LTD., Crymych, UK, with a 0.2 mm precision and measurement range from 0 to 48 mm; a bicondylar pachymeter to measure the diameter of small bones, model Holtain, with 1 mm precision and measuring range from 0 to 140 mm; and a dermographic pencil to mark anatomical points. The collected variables were body weight, waist, and hip circumference, as well as triceps, subscapular, supraspinal and abdominal folds. Measurements were taken twice, with a third measurement taken in the event that the difference between the first two measurements was greater than 5% for the folds and 1% for the other measurements. 

In order to measure body weight (kg), we requested participants to wear light clothes, take their shoes off, and stand up straight in the center of the scale, looking forward, with upper limbs resting on both sides of the body and trying to distribute weight equally on both feet. Height (cm) was measured without them wearing shoes, with the subject standing on both feet and with feet together, with heels forming a 45° angle, ensuring that the head was in a Frankfort plane position. Waist circumference (cm) was taken at the point between the bottom part of the last rib and the highest part of the hip, at the end of a normal exhalation and with arms relaxed on either side of the body. Hip circumference was taken with the subject standing up, with legs and knees together and relaxed glutes. The anthropometrist placed the anthropometric tape at the maximum circumference of the glutes, perpendicular to the longitudinal axis of the torso. The necessary anatomical points were marked with a dermographic pencil before taking skinfold measurements. All skinfolds (mm) were taken from the participant’s right side. The waist-to-hip ratio (WHR), which is an anthropometric measurement to measure intra-abdominal fat levels, was determined using the ratio of waist circumference to that of the hip [35]. 

Beck Depression Inventory II (BDI-II): BDI-II was validated in the general population in 2003, showing greater internal consistency and factorial validity than its predecessor, BDI-IA. BDI-II has undergone some modifications compared to previous versions in order to better represent the criteria to diagnose depressive disorders included in DSM5. The test is preferably intended to be used as a clinical tool as a means of assessing the severity of depression in adults and teenagers over 13 years of age [36].

This was a questionnaire that assessed symptoms of depression in patients, mainly of a cognitive type, although physiological, emotional or motivational types were also evaluated. This version also included symptoms such as agitation, feelings of worthlessness, difficulty in concentrating and loss of energy. Each item on the questionnaire reflected a symptom of depression, and each one offered four alternative statements, ordered from least to most serious. The assessed person had to choose the phrase from each of the 21 sets of four options that best reflected how they had felt over the last week, including the day they fill out the questionnaire. Each item was assessed from 0 to 3 points, according to the chosen option and, after directly adding the score for each item, a total score was obtained, ranging from 0 to 63, that quantified the presence and severity of symptoms of depression [37].

Blood analysis: A blood sample was taken at 9 a.m. on an empty stomach. After the test, the serum was then separated from the plasma after centrifuging the samples. Cortisol was measured by means of a chemiluminescent immunoassay using commercial kits (Immulite). Concentrations of BHB and albumin were measured with commercial kits (Randox for BHB and Beckman for albumin) in an automatic clinical biochemical analyzer (Olympus A 400).

### 2.6. Ethical Concerns

This study was carried out in accordance with the Helsinki Declaration [38], with prior approval of the protocol by the Human Research Committee of the University of Valencia of the Experimental Research Ethics Committee (procedure number H1512345043343). In addition, patients involved in the study signed a consent form after being informed of the procedures and nature of the study.

## 3. Results

Once the selection criteria were applied, the population sample of 51 patients with MS was divided into the intervention group and the control group. The sociodemographic and clinical characteristics of both groups are shown in Table 1, where it can be seen that there were no significant differences in the analyzed variables between groups.

After 4 months of treatment,significant increases in BHB and albumin, and a decrease in depression, were observed in the intervention group (Table 2). There were no significant changes in the cortisol levels in the blood in any group. 

Regarding anthropometric changes, significant decreases in WHR, fat percentage and fat weight occurred in the intervention group, while no changes were observed in the control group. Furthermore, an important rise in muscle percentage was observed in the intervention group, yet it decreased considerably in the control group. Finally, in terms of total weight, a significant decrease was detected in both groups. It should be noted that no significant differences were observed between male and female in the total weight, RCT and fat variables, neither in the control nor intervention group, and neither before nor after the treatment (*p* > 0.05); solely, a difference between sexes was observed in pre-treatment in the control group in the variable RCT (*p* = 0.03) (Table 3).

These results, together with the significant increase in albumin protein already indicated above, could determine a decrease in the concentrations of free cortisol in the intervention group.

Finally, it should be noted that, throughout the 4-month intervention, no relapses of the disease were registered in any study participant.

## 4. Discussion

MS is characterized by a mitochondrial alteration of the neurons as a result of high levels of oxidative stress and, at the same time, is related to the presence of depression [39]. It is also clinically related to obesity and neuromuscular deterioration [40]. 

There are few studies that have researched possible emotional improvement after non-pharmacological interventions in MS patients. This is mainly due to the complexity of the etiopathogenesis of depression involving several factors (genetic, biological and psychosocial). Among these factors, the hyperactivation of the HHA axis with high levels of cortisol, related to the presence of depression in certain pathological situations, stands out [41]. In this sense, patients with MS do have an HPA axis hyperactivity, with the consequent increase of ACTH and cortisol levels in the blood [7]. EGCG can reduce HPA axis dysfunction [42]. This could explain its impact on improving depression levels: in patients with symptoms of anxiety and depression related to schizophrenia or bipolar disorder [24], these symptoms were reduced in in animal models after administering catechin for 8 weeks [25]. These improvements would match with our results, since we observed a decrease in depression only in the intervention group. However, it does not seem that this change is due to a decrease in levels of total cortisol in blood, since, as observed in the control group, its levels remain similar after treatment. This could be due to the fact that, matching with these results, it has been evidenced that the administration of EGCG does not decrease levels of cortisol in blood [43], even though it does inhibit 11β-HSD1. This enzyme is responsible for the passage from cortisone to active cortisol, which is the one that finally binds to its receptor, producing the harmful effect [44]. Moreover, it is interesting to highlight the role of albumin in free cortisol activity, since, in situations of hypoproteinemia, lower levels of cortisol transport are observed, increasing its free form even though its total levels remain the same [45]. It could be postulated that in our study the increase in albumin can lead to a decrease in therefore the amounts of free cortisol which can bind to its receptor (GR). However further studies should be made to confirm this mechanisms.

The high concentrations of albumin and decrease in body fat mass could be related to a decrease in inflammation. Albumin is a negative acute phase protein that decreases in inflammation [46], and whose low levels are associated with greater percentages of body fat and higher concentrations of IL-6 and CRP in the blood [47]. In addition e, excess body fat mass promotes inflammation, which aggravates the prognosis of the disease and, in turn, is related to the presence of depression in MS patients [12]. Furthermore, an especially significant relation between depression and fat has been established when the latter is located around the abdomen [12,48,49]. Thus, the obtained reduction in fat in our study, mainly around the abdomen, could help to explain improvements in the levels of depression in the intervention group through a drop in inflammation as a result of anthropometric improvements. We should outline that our results coincide with those obtained in other studies after administering EGCG [50,51]. It is important to point out that, there is a relation between negative emotional symptoms and MS. 

In our study, ketone bodies were significantly increased) in the intervention group after administering 60 mL daily. Ketogenic diets, and especially the consumption of coconut oil, due to its high percentages in medium-chain fatty acids that exceed 50% (49% lauric acid, 8% caprylic acid, 7% capric acid and a low percentage of caproic acid) [52], are the most important source of ketone bodies after hepatic metabolism. Medium-chain fatty acids have a high level of oxidation for obtaining energy, thus avoiding storage in adipose tissue and promoting higher energy use [53] that leads to weight loss without recovery in the long term [54]. This lipolytic effect of ketone bodies has been associated with improvements in depression [55,56,57,58]. In this line, anxiety and depressive symptomatology stand out among these types of patient [59,60,61]. This is why the study of possible associated factors with the expression of this symptomatology in MS patients becomes essential—to practically improve these patients’ quality of life. 

The side effects of the drugs used to treat these types of patient can produce negative effects. Therefore, we suggest that the development of non-pharmacologic treatments based on alternative therapies, such as the administration of antioxidants, effective for practical effects in MS patients, as our results show, is essential. Being able to know the beneficial impact of these possible treatments and understanding how they achieve these improvements, as we have tried to do in our work, will allow us to advance knowledge in the area, improving the quality of life of these patients. All the aforementioned information makes our results promising for understanding and treating MS.

Despite these findings, our study has limitations that should be taken into account for future projects. For example studies should be performed about the possible correlation betweenthe anthropometric changes and biomarkesr or imaging test that can indicate the probable energy improvements on a mitochondrial level. It would also be of interest the evaluation of the concentrations of albumin, cortisol and even BHB, not only before and after 4 months of intervention, but also throughout the intervention. Furthermore, studies regarding changes in ACTH and their relation with depression should be performed.

## 5. Conclusions

The combination of EGCG and coconut oil as a source of ketone bodies in the blood is associated with an increase in albumin in blood, a decrease in abdominal fat, and lower depression. 

## Figures and Tables

**Figure 1 life-11-00353-f001:**
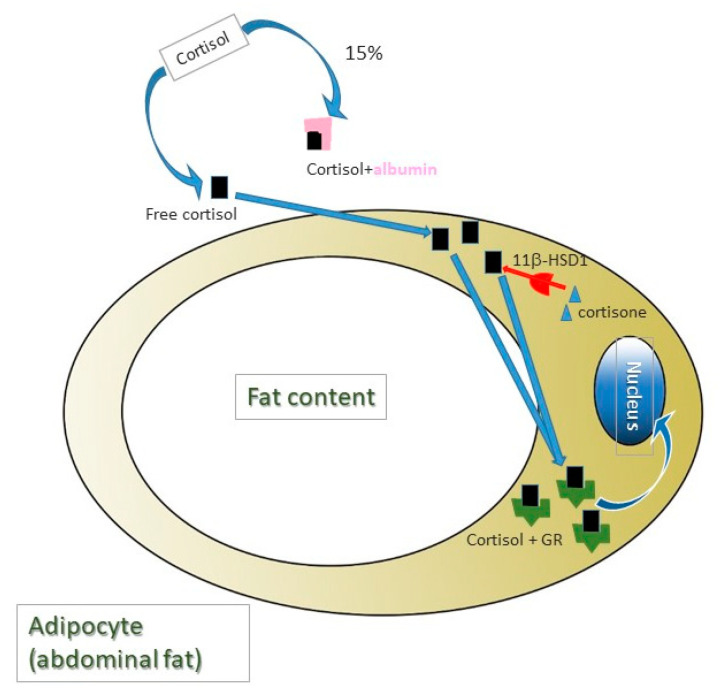
Cortisol is a lipidic nature molecule and require carrier proteins for its blood transport. Albumin transports 14% of total cortisol and 6% of total cortisol in blood is in free form. Free cortisol quantity is correlated positively with body fat mass, mainly from abdominal region as a consequence of a higher expression of 11β-hydroxysteroid dehydrogenase type 1 (11β-HSD1), which turns cortisone into cortisol. Cortisol molecules (not cortisone), in turn, bind to their receptor (GR), which, once activated, is translocated to the nucleus, where it acts as a transcription factor.

**Figure 2 life-11-00353-f002:**
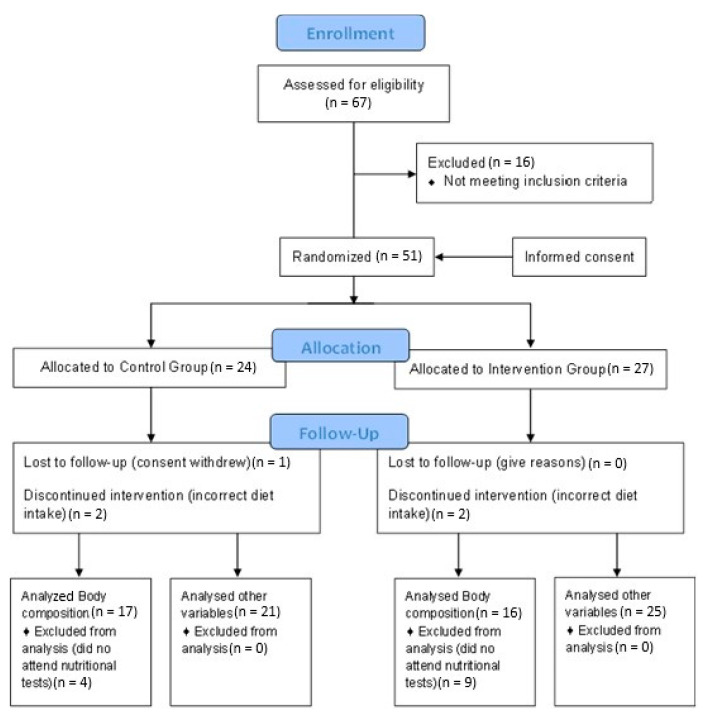
Consort Flow Diagram.

**Table 1 life-11-00353-t001:** Sociodemographic and clinical characteristics of the population of the study.

		Control Group N = 24		Intervention Group N = 27		Chi2	*p*
		Count	%	Count	%		
Relapsing-remitting MS		17	70.8%	20	74.1%	0.067	0.796
Secondary-progressive MS		7	29.2%	7	25.9%		
Men		10	41.7%	5	18.5%	3.279	0.070
Women		14	58.3%	22	81.5%		
		**Mean**	**SD**	**Mean**	**SD**	**Z**	***p***
Age (years)		49.83	12.42	44.56	11.27	−1.558	0.119
Time since diagnosis (years)		14.21	8.40	11.89	9.74	−1.418	0.156
Pre-test	Albumin (g/dL)	4.66	0.41	4.69	0.29	−0.414	0.679
	Cortisol (µg/dL)	15.50	5.86	12.22	4.59	−1.930	0.054
	BHB (Mmol/L)	0.05	0.02	0.06	0.04	−0.932	0.351
	Depression	11.17	8.66	13.59	8.70	−1.096	0.273
	Waist-to-hip ratio	0.95	0.08	0.89	0.10	−1.625	0.104
	Body fat mass (%)	18.85	5.00	19.53	3.78	−0.764	0.445
	Fat weight	14.27	7.47	13.65	5.26	−0.311	0.755
	Muscle mass (%)	38.38	4.15	39.39	2.88	−0.547	0.584
	Weight (kg)	70.44	18.13	68.63	13.56	−0.245	0.806

Chi2: Chi square test; Z: Mann–Whitney U test; MS: multiple sclerosis; BHB: beta-hydroxybutyrate; SD: standard deviation.

**Table 2 life-11-00353-t002:** Intra-group comparison before treatment (pre) and after treatment (post) in the quantified scalar variables.

Control Group	Pre (N = 21)		Post (N = 21)			
	Median	Range	Median	Range	Z	*p*
Cortisol (µg/dl)	16.30	23.85	13.10	29.36	−1.731	0.884
Albumin (g/dL)	4.65	1.43	4.69	1.80	−1.469	0.142
Depression	9.50	34.00	9.00	27.00	−2.508	0.072
BHB (Mmol/L)	0.04	0.06	0.03	0.17	−1.254	0.210
	**Pre (N = 25)**		**Post (N = 25)**			
**Intervention Group**	**Median**	**Range**	**Median**	**Range**	**Z**	***p***
Cortisol (µg/dL)	11.90	18.44	11.05	18.01	−0.317	0.751
Albumin (g/dL)	4.65	1.14	4.81	0.86	−2.375	0.018 *
Depression	12.00	28.00	8.00	24.00	−2.704	0.007 *
BHB (Mmol/L)	0.05	0.16	0.05	0.33	−2.005	0.045 *

BHB: beta-hydroxybutyrate; SD: standard deviation; *: statistically significant differences *p* < 0.05; Z: Wilcoxon signed-rank test.

**Table 3 life-11-00353-t003:** Intra-group comparison before treatment (pre) and after treatment (post) in the variables related to levels of fat, muscle, and total weight.

Control Group	Pre (N = 17)		Post (N = 17)			
	Median	Range	Median	Range	Z	*p*
WHR	0.93	0.39	0.94	0.40	−0.492	0.623
Body fat mass (%)	18.19	16.19	17.72	19.22	−0.644	0.520
Fat weight	12.61	26.77	12.71	28.10	−0.355 ^c^	0.723
Muscle mass (%)	38.88	16.30	39.10	17.69	−1.738	0.082
Weight (kg)	62.45	68.60	61.00	63.00	−2.723	0.006 *
	**Pre (N = 16)**		**Post (N = 16)**			
**Intervention Group**	**Median**	**Range**	**Median**	**Range**	**Z**	***p***
WHR	0.90	0.35	0.88	0.30	−2.183	0.029 *
Body fat mass (%)	19.40	14.50	17.63	13.34	−4.421	0.000 ***
Fat weight	12.95	20.85	11.17	18.51	−4.373	0.000 ***
Muscle mass (%)	39.77	12.68	40.68	11.97	−2.955	0.003 **
Weight (kg)	64.50	56.10	65.50	53.00	−3.076	0.002 **

WHR: waist-to-hip ratio; SD: standard deviation *: statistically significant differences *p* < 0.05; **: statistically significant differences *p* < 0.005; ***: statistically significant differences *p* < 0.001; Z: Wilcoxon signed-rank test.

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
