# Peer review of "The Impact of Epigallocatechin Gallate and Coconut Oil Treatment on Cortisol Activity and Depression in Multiple Sclerosis Patients"

_life, 2021, doi:10.3390/life11040353_

Round 1
Reviewer 1 Report
There is an interesting paper, however, some improvement is required.
Changes in WHR and fat mas are presented for the whole group. There are significant differences in fat distribution between men and women. Did you see differences between men and women?
The observed changes in depression scale seems to result from both EGCG supplementation and ketogenic diet as increase in BHB levels was observed. Therefore, an improvement under results and discussion is needed.
In conclusion, decrease is cortisol is underline, however, no significant changes in cortisol under treatment was observed. Therefore, an improvement is needed.
Reviewer 2 Report
The article shows possibly interesting and valuable results, but it requires certain improvements.
FIRSTLY: A CONSORT checklist, with appropriate adjustments in the manuscript, should be included.
The introduction is chaotic, imbalanced and does not really provide rationale for the current study. The antioxidant theory is not the only concept trying to explain the pathophysiology of MS - this probably should be underlined. Then, there is a rather detailed description on the transporting of cortisol, with a Figure presented (is it a figure devised by the authors) and then suddenly - a mention on the efficacy of treatment and then on possible role of antioxidant agents - with quite bold statements. Consider also the following:
Page 2 line 50 - the statement that „the role of cortisol in onset of depression is unclear” is imprecise - does it refer solely to MS patients or general population? Te reference is regarding MS patients, yet I would suggest a bit wider mention on possible role of cortisol in ethiology/mechanisms of depression.
Page 2 Line 53 - „Furthermore, chronic high levels of cortisol, have been associated with depression through abdominal fat presence” - this is not the only mechanism, the statement is very general, yet does not reflect the knowledge on the association between cortisol concentration and depression.
Materials and Methods:
Subjects - please specify the reason for exclusion of the 16 patients before randomization - either in the text or in the CONSORT diagram.
Statistical analysis (and thus Results) - since the distribution was non-normal consider presenting median (with quartile1 - quartile 3 range) instead of mean +/- standard dev
Results
Page 6 line 223-226 - those statements are not true, imprecise. It should be rather stated that the differences in terms of the mentioned variables between the groups were statistically insignificant. And thus - it may be stated that the groups were similar regarding the variables of interest. (Please note that the reported mean values does not necessarily back the statement of no differences. Also - I refer you to to basic knowledge on statistical testing - they do not provide us rationale on stating on lack of difference).
Wilcoxon tests operates only on paired variables. Thus, in Table 3 there is an error - the table should contain only data for those who completed the trial, also - only then it makes sense. This is a major methodological error that raise my concern regarding the quality of the whole research and correctness of the statistical analysis and reasoning. Please verify and improve this part - this is essential for consideration for publication.
The Discussion deliver very sharp explanations for possible mechanisms of the observed effects. I think that postulation of possible mechanism, with a Figure added - is a bit too much. I would suggest the authors to focus on possible practical application on the results, rather than elaborated speculations on possible mechanisms. The current study does not provide much of rationale for detailed pathophysiology descriptions.
Limitations belong in the discussion, not conclusions.
Round 2
Reviewer 2 Report
The authors provided improvements in accordance with my suggestions. Thus, the article qualified for publication in the current form,